# FPI Based Hyperspectral Imager for the Complex Surfaces—Calibration, Illumination and Applications

**DOI:** 10.3390/s22093420

**Published:** 2022-04-29

**Authors:** Anna-Maria Raita-Hakola, Leevi Annala, Vivian Lindholm, Roberts Trops, Antti Näsilä, Heikki Saari, Annamari Ranki, Ilkka Pölönen

**Affiliations:** 1Faculty of Information Technology, University of Jyväskylä, 40100 Jyväskylä, Finland; leevi.a.annala@jyu.fi (L.A.); ilkka.polonen@jyu.fi (I.P.); 2Department of Dermatology and Allergology, University of Helsinki and Helsinki University Hospital, 00290 Helsinki, Finland; vivian.lindholm@hus.fi (V.L.); annamari.ranki@hus.fi (A.R.); 3VTT Technical Research Centre of Finland Ltd., 02150 Espoo, Finland; roberts.trops@vtt.fi (R.T.); antti.nasila@vtt.fi (A.N.); heikki.saari@vtt.fi (H.S.)

**Keywords:** hyperspectral, FPI, calibration, interferometry, optical modelling, convolutional neural network, LED illumination, photometric stereo, skin surface model, biomedical imaging, dermatological application, optical biopsy

## Abstract

Hyperspectral imaging (HSI) applications for biomedical imaging and dermatological applications have been recently under research interest. Medical HSI applications are non-invasive methods with high spatial and spectral resolution. HS imaging can be used to delineate malignant tumours, detect invasions, and classify lesion types. Typical challenges of these applications relate to complex skin surfaces, leaving some skin areas unreachable. In this study, we introduce a novel spectral imaging concept and conduct a clinical pre-test, the findings of which can be used to develop the concept towards a clinical application. The SICSURFIS spectral imager concept combines a piezo-actuated Fabry–Pérot interferometer (FPI) based hyperspectral imager, a specially designed LED module and several sizes of stray light protection cones for reaching and adapting to the complex skin surfaces. The imager is designed for the needs of photometric stereo imaging for providing the skin surface models (3D) for each captured wavelength. The captured HS images contained 33 selected wavelengths (ranging from 477 nm to 891 nm), which were captured simultaneously with accordingly selected LEDs and three specific angles of light. The pre-test results show that the data collected with the new SICSURFIS imager enable the use of the spectral and spatial domains with surface model information. The imager can reach complex skin surfaces. Healthy skin, basal cell carcinomas and intradermal nevi lesions were classified and delineated pixel-wise with promising results, but further studies are needed. The results were obtained with a convolutional neural network.

## 1. Introduction

Hyperspectral imaging (HSI) systems can be utilized in various sensing applications. The highly dimensional hyperspectral (HS) data offers high accuracy and robustness for identification and characterisation tasks [1,2]. A HS image can be considered a stack of frames, each representing the intensity of a different wavelength of light. Since each spatial pixel has a spectrum, the HS image contains spatial and spectral domains, which enables the accurate pixel-wise classification [3].

Spectral imaging systems were originally applied in remote sensing applications, where systems are robust to rough or irregular topographies, because imaging is performed from the satellites or airplanes. When imaging is performed from a closer range, surface topography and tomography start to affect the image’s spectral quality and homogeneity. Recent advance in hyperspectral sensor imaging has made sensors smaller [4]. Using the Fabry–Pérot interferometer (FPI) as a spectral separator, the imager size can be reduced without bargaining from spatial or spectral resolution.

HS imaging is proven to be a powerful tool for detecting and identifying diseases in medical research [5,6,7]. In biomedical imaging, one potential area is dermatological applications. With hyperspectral imaging systems it is possible to delineate malignant tumors [8,9,10], detect invasions [11,12,13,14] and even classify lesion types [15]. These studies pointed out that complex surface topography and tomography is one major challenge when shoulders, nose, chin or other facial skin areas are imaged with hyperspectral cameras. Hyperspectral imaging of such complex areas requires a small-sized hand-held spectral imager. Additionally, the illumination needs to be handled similarly, as the illumination of the hyperspectral images needs to be stable and constant in order to the captured data to be easily analysed and processed.

This article is the first stage of a pilot study completed in three stages from 2020 to 2022. The aim is to introduce the working principles and clinically pre-test the concept of the new, compact hand-held SICSURFIS Piezo-actuated metallic mirror FPI hyperspectral imager (SICSURFIS HSI) for complex skin surfaces. This pre-test enables the system engineering, imaging, and analysis aspects of further system development toward a clinical application.

The SICSURFIS HSI is designed to face the mentioned challenges of the previous studies. For illumination and challenging skin surfaces, there is a controllable built-in light-emitting diode (LED) based illumination module, which is designed for photometric stereo imaging. The special stray light protection cones will block the unwanted light and enable the imager to reach and adapt to complex skin surfaces. The imager produces spectral, spatial and surface topography information.

In the data processing, the raw measurement data were processed into the surface albedo, normals and surface model. The processing was performed with common hyperspectral reflectance calculations and a photometric stereo method for the surface model calculation [16]. Training and testing data sets were composed of windowed pixels (HS-sub-cubes) from the lesions that were histologically confirmed by an experienced dermatopathologist. This pixel-wise classification approach provides large training data (31,168 HS-sub-cubes after augmentation) for the machine learning model.

Our hypothesis is that these surface models can improve the hyperspectral imaging machine learning model’s performance in clinical decision making and the new imaging concept can adapt to the complex skin surfaces. The skin surface models combined with the spectral and spatial domains will benefit the classification and delineation results of the convolutional neural network (CNN). In this first-stage pilot study, the aim is to compare the capacity of the HS in differentiating malignant basal cell carcinomas (BCCs) from clinically similar-appearing but benign intradermal nevi (ID) by comparing the surface models and albedo maps of the measured BCCs and IDs with CNN. So far, this is the pre-study for testing the concept, and evaluating the technical and instrumental aspects from the analysis point of views. The prototype imager with its application is not ready for a clinical use, but the first results and future potential can be evaluated. The mentioned second and third study stages will be independent continuation clinical pre-studies, with more lesions and lesion types. The imaging procedure, methods, and machine learning models will be improved based on these first steps.

The paper is organised as follows. Section 2 describes the overall and mechanical design and system validation of a novel skin cancer HS imager. Section 3 describes the steps of a clinical pre-test from instrumentation (Section 3.2), materials and methods (Section 3.3) to the results (Section 3.4) of the photometric stereo and the spectral 3D classification. The discussion (Section 4) and conclusions (Section 5) finalise the study.

## 2. The Sicsurfis Hyperspectral Imager Concept

This section introduces the SICSURFIS HSI’s concept in six subsections; the design and operating principles, spectral responses and imaging calibration are presented in Section 2.1 and Section 2.2. The following topics are LED illumination module (Section 2.3), optical design (Section 2.4) and system validation (Section 2.5). The photometric stereo imaging methods (Section 2.6) will finalise the concept presentation.

### 2.1. The Design and Operating Principles of the Spectral Imager

The SICSURFIS Spectral Imager (SICSURFIS HSI) was designed to provide spectral images enlightened from different angles of light for the photometric stereo algorithm. The stray light protection cones (Figure 1) block the unwanted background light and ensure the correct distance for focusing the images. The soft silicone collars adapt the stray light protection cones softly to complex skin surfaces and provide comfort for the patient, since the imager must be held relatively tightly against the skin. The silicone collar is resistant to strong surgical disinfectants. The diameters of the light protection cones are 2.0, 3.6, 4.9 and 8.5 cm.

SICSURFIS HSI’s modules, sensor, spectral separator and LEDs are independently controllable, which enables effective configuration via software. Imagers spectral range is 475–975 nm and the spectral resolution (full-width at half maximum, FWHM) 8–18 nm. The average FWHM in selected spectral channels in this study was ∼10 nm. In this study, we utilised 33 selected spectral channels, but the imager is, depending on the calibration, capable of capturing hundreds or even thousands of spectral channels. The imager’s pixel resolution is approximately 24 μm× 24 μm. In spectral imaging, the tissue penetration depth depends on the wavelength. In this study, the used wavelengths had penetration depth from 0 to 6 mm [17].

The SICSURFIS HSI with its LED module and stray light protection cones is a small-sized hand-held device, the overall weight of the imager is 880 g. The imager is easy to apply to patients’ skin and it reaches the challenging places. The patient does not have to move, so imaging is also possible in patients with reduced mobility. The imager is a spectral scanner, but it is almost as fast as a snapshot. Nor does it describe hundreds of wavelength ranges and thus add extra dimensions to the data. The number of wavelengths and the wavelengths of interest can be selected on a case-by-case basis, just as the LEDs can be individually controlled to match the current wavelength. This study used the wavelengths that had spectral absorption peaks in tissue chromophores such as melanin, haemoglobin, water, beta-carotene, collagen, and bilirubin [18].

The size and adaptivity are advances compared to devices used in previous studies. Example images of previous and current HSI systems can be seen in these articles [13,14,19,20,21,22], (Figure 1 in all of them). The SICSURFIS HSI’s imaging setup can be seen in Section 3.2, Figure 11a.

The imager consists of a Piezo-actuated metallic mirror Fabry–Pérot Interferometer (FPI), an RGB sensor and an LED light source. The FPI controls the light’s transmission to the RGB sensor, and the role of the separate long and short pass filters, shown in Figure 2, is to cut the unwanted transmission at not selected orders of the FPI. The sensor’s basic principle is to provide different spectral layers by changing the FPI air gap [4]. Typically the FPI air gap can be changed to a new value in less than 15 ms, i.e., the settling time of the air gap is 15 ms.

Those FPI orders are matched with different sensitivities of the image sensor channels. There are three wavelength channels with different pixels in the RGB sensor’s Bayer pattern. If we carefully select the FPI air gap range, the RGB sensor will receive one to three transmission peaks. After recording the transmission peaks, the different spectral responses of the red, green and blue pixels can be seen throughout the selected spectral range [4,23].

### 2.2. Pixel Spectral Response and Imager Calibration

Before spectral calibration, we demosaic each Bayer pattern frame using bilinear interpolation, so that response in each pixel is now vector $s\in R^{3}$. We can now arrange these pixels to a response matrix S, which is [3×n] matrix, where *n* is the number of pixels. We are interested in reconstructing radiance R which is [n×3] matrix. Described FPI system makes it possible to have one to three wavebands with one FPI gap. We need to determine the function between these two values to achieve this. The separate spectral calibration process in Section 2.5.1 will give use FPI gap wise coefficient matrix C([3×3] matrix) using radiance $R_{\lambda }$ information from the reference spectrometer so that
(1)$$S=CR_{\lambda }.$$ From Equation (Equation 1) we solve coefficient matrix *C*. Now, the radiance is:(2)$$R=C^{-1}S.$$

### 2.3. Led Illumination System

The LED illumination system of the SICSURFIS HSI is designed for the photometric stereo imaging setting. The LED light source module is a driver electronics PCB with 27 LEDs. The inner radius of the ring is 33 mm. The LED light source module has 27 pieces of intentionally selected 5 mm LEDs: white, 680, 720, 750, 780, 810, 850, 880 and 940 nm. Three series of these 9 LEDs are tilted by an angle of 30 degrees relative to the system optical axis. LEDs and FPI positions are individually controllable. The captured wavelengths, selected with FPI, can be combined with according LED illumination via software. Figure 3 visualizes the LED light source module’s technical details. The temporal stability of the LED light source is taken into account by keeping the system on for several minutes before the recording of spectra. The other way to control the intensity stability is to record white reference images frequently.

### 2.4. Optical Design

The optics of the SICSURFIS HSI is designed with commercial S-mount and C-mount lenses, which provide collimated light beam through the Piezo-actuated FPI tunable filter. The selected optical design is described in Figure 4. The Lensagon CMFA0420ND C-Mount lens (F-number 2.0, focal length = 4.16 mm) forms an intermediate image at the focal plane of the Lensagon B5M16020V2 5 Megapixel S-Mount lens (F-number = 2.0, focal length = 16 mm). This lens collimates the light coming from the intermediate image. The collimated light goes through the Piezo-actuated FPI tunable filter to the Lensagon B5M25024V2 5 Megapixel S-Mount lens (F-number = 2.4, focal length = 25 mm), which focuses the image of the target on the image sensor.

### 2.5. System Validation

#### 2.5.1. The Monochromator Calibration Setup

Before the monochromator calibration, the transmission spectra characterization of the selected FPI module was carried out. The spectra were measured with the Ocean Optics HR4000 spectrometer at the center of the FPI. The monochromator calibration of the SICSURFIS HSI camera was performed using the setup shown in Figure 5.

The combined sensitivities of the FPI and red, green and blue pixels were determined for over 200 FPI air gaps. In the calibration data analysis, the coefficients for the combined sensitivities of the FPI and red, green and blue pixels are retrieved in such a way that the linear combination of the R-, G- and B-pixel sensitivity signals contains none zero signal only at one spectral band.

In the monochromator calibration, the signals of the calibrated optical power meter (Thorlabs PM16-120 USB Power meter) and the SICSURFIS HSI were recorded for the wavelength range 475–975 nm at 2 nm intervals and the Full-Width Half Maximum (FWHM) resolution of 5 nm.

The results of the system calibration for SICSURFIS HSI are the raw measurement data of the monochromator calibration, the spectral response functions of R-, G- and B pixels for Piezo-actuated FPI (PFPI) setpoint voltages used in the calibration, and the system calibration matrices that can be used to calculate the spectral radiance or photons at the input aperture of the hyperspectral camera optics from the raw pixel signals. The exposure time and the camera gain for pixels must be known relative to the exposure time and pixel gains used in the monochromator calibration in order for absolute calibration to be valid. Examples of the measured combined sensitivities of the FPI and red, green and blue pixels and linear combinations of these sensitivities are presented in the next section.

#### 2.5.2. Led Illumination System

The LED light source module’s spectral radiances were measured using calibrated fiber, HR4000 spectrometer and white balance reflectance target, using the setup shown in Figure 6. The LEDs were selected to cover the wavelength range of the used FPI. The results of the single LED spectral radiance measurements can be seen in Figure 7.

#### 2.5.3. Sicsurfis Hsi Camera Monochromator Calibration Pixel Sensitivity Function Results

The purpose of the monochromator calibration is to determine the combined sensitivities of the FPI and red, green and blue pixels (see Figure 8). The peak wavelengths 1, 2 and 3 are the determined by the spectral transmission spectrum of the FPI (lower left part of Figure 8). When the FPI transmission curve and R-pixel quantum efficiency are multiplied, we get the Red curve in the center right part of Figure 8. Similarly, we get the Green curve for G-pixels and the Blue curve for the B-pixels in the center right part of Figure 8. An example of the results of the SICSURFIS HSI camera monochromator calibration are shown in Figure 8, Figure 9 and Figure 10.

In Figure 9, the scaled R-, G- and B-pixel sensitivity signals are plotted in units DN/(W/nm). The linear combination of the scaled pixel sensitivity functions for the peak wavelengths 1, 2 and 3 can be seen in Figure 10. There are two peak wavelengths for the selected PFPI drive voltage; 548.2 nm (on the left) and 812.2 nm (at the center).

### 2.6. Photometric Stereo Imaging

Photometric stereo is an imaging setting where the imaged object is illuminated from multiple angles, and corresponding images are captured with a stationary camera. When the angles of the lights are known compared to the imaged object, one can calculate the surface normals of the imaged object based on the intensity of gathered light in different imaging angles [24]. From the surface normals, one can calculate a three-dimensional surface model with the Frankot–Chellappa algorithm [16].

The normal maps for each wavelength are calculated by multiplying the inverse matrix of the light direction matrix by the three reflectance values for each pixel, corresponding to each light direction. The light directions are defined by the length from the imaged object to the center of the LED-module, 65±5mm, the diameter of the LED-module, 33 mm, and the placement of the LEDs on the module, as described in Figure 3. Same LEDs are separated by 120 degrees, i.e., $\frac{2\pi }{3}$, on the edge of the module, and the angle between the object layer and the arriving light vector is 60 degrees, $\frac{\pi }{3}$. This gives us the normalized light direction matrix for the center of the imaging field:(3)$$L=\frac{1}{2}\left[\begin{array}{ccc} 0 & 1 & \sqrt{3}\\ \frac{\sqrt{3}}{2} & -\frac{1}{2} & \sqrt{3}\\ -\frac{\sqrt{3}}{2} & -\frac{1}{2} & \sqrt{3} \end{array}\right].$$

Now the normal *N* and albedo *a* matrices for each wavelength band are calculated as
(4)$$aN=L^{-1}\;\cdot \;\left[\begin{array}{ccc} r_{000} & r_{001} & r_{002}\\ r_{010} & r_{011} & r_{012}\\ \vdots & \vdots & \vdots \\ r_{km0} & r_{km1} & r_{km2} \end{array}\right],$$
where each subscript for reflectance *r* mean its *x*-coordinate, *y*-coordinate, and the used light direction, respectively. The resulting matrix, which is of shape k·m rows and three columns, is then reorganized into the normal map by transforming it into the original image shape:(5)$$aN_{\mathrm{map}}=\left[\begin{array}{cccc} a_{00}\left(n_{000},n_{001},n_{002}\right) & a_{01}\left(n_{010},n_{011},n_{012}\right) & \ldots & a_{0m}\left(n_{0m0},n_{0m1},n_{0m2}\right)\\ \vdots & \vdots & \ddots & \vdots \\ a_{k0}\left(n_{k00},n_{k01},n_{k02}\right) & a_{k1}\left(n_{k10},n_{k11},n_{k12}\right) & \ldots & a_{km}\left(n_{km0},n_{km1},n_{km2}\right) \end{array}\right],$$
where albedo $a_{ij}$ are the lengths of the vectors:(6)$$aij=|(aN_{ij0},aN_{ij1},aN_{ij2})|.$$

From this, we can calculate the partial derivatives of the depth in *x*- and *y*-direction in every point (*a*,*b*) and denote them by *p* and *q*:(7)$$\begin{array}{cc} p_{ab} & =\frac{\partial z}{\partial x}=\frac{n_{ab0}}{n_{ab2}}, \end{array}$$(8)$$\begin{array}{cc} q_{ab} & =\frac{\partial z}{\partial y}=\frac{n_{ab1}}{n_{ab2}}. \end{array}$$

Now we calculate the Fourier transform F for each *p* and *q* map:(9)$$\begin{array}{cc} P & =F(p), \end{array}$$(10)$$\begin{array}{cc} Q & =F(q). \end{array}$$

Now the estimated surface map is:(11)$$Z=R\left(F^{-1}\left(\frac{-i\omega_{x}\;\cdot \;P-i\omega_{y}\;\cdot \;Q}{\omega_{x}^{2}+\omega_{y}^{2}+\epsilon }\right)\right),$$
where R represents the real part of the value, $F^{-1}$ the inverse Fourier transform and ω the frequencies in the Fourier transform. ϵ is added to avoid division by zero. The details of the method can be found in [16].

## 3. Clinical Pre-Test: Surface Model Classification of Basal Cell Carcinomas and Intradermal Nevi

The clinical pre-test section is divided into four sections. The background (Section 3.1) and instrumentation (Section 3.2) leads to the material and methods (Section 3.3), which carefully explain the steps from spectral data and pre-processing (Section 3.3.1) to machine learning pre-processing (Section 3.3.2), method validation (Section 3.3.3) and lesion classification (Section 3.3.4). After the methods, we will present the results in Section 3.4; the photometric stereo and albedo spectra (Section 3.4.1) and pixel-wise classification (Section 3.4.2) with the slice half model and the leave-one-out validation (Section 3.4.3).

### 3.1. Background

The clinical data gathering was performed in two phases during the spring and autumn of 2020. For the demonstrative pilot-test in this study, we examined two types of skin lesions, malignant basal cell carcinomas (BCC) and benign intradermal nevi (ID), with a clinically similar appearance.

The HS images were captured in the first phase, and the lesions were clinically diagnosed and annotated by dermatologists at Helsinki University Hospital. In the second phase, we calculated the surface models of the skin surfaces using the methods of photometric stereo imaging. The aim was to classify the pixel spectra of BCCs, IDs, and healthy skin. We used a convolutional neural network and pixels selected from the albedo images and the lesions’ depth data.

We captured images of 14 BCCs and 8 IDs on 21 volunteering patients with HS, digital and dermoscopic imaging and subsequently removed the lesions for dermatohistopathological analyses to confirm their diagnoses. All volunteering patients provided their written informed consent. The study protocol followed the Declaration of Helsinki and was approved by the Ethics Committee of Helsinki University Hospital.

One of the typical challenges related to CNN classification and HS imaging is the limited availability of the labelled training data, which can lead the models to overfit [3]. Thus we selected a pixel-wise classification approach. The training data set consisted of 6160 windowed HS-sub-cubes collected from the HS images of histologically confirmed skin lesions. After data augmentation training set contained 31,168 HS-sub-cubes.

### 3.2. Clinical Pre-Test Instrumentation

The HS images were captured by dermatologists and nurses (users). The SICSURFIS HSI was combined with a specially designed hospital version of the CubeView software [25]. CubeView is a spectral imaging and analysis software developed by the spectral imaging laboratory of the University of Jyväskylä, Finland [26]. It controls the SICSURFIS HSI’s machine vision sensor, PFPI and LED modules by using the Camazing [27] and the Spectracular [28] Python libraries.

The data capturing setup was designed for effortless workflow. All of the device and setup settings (e.g., exposure time, LEDs, wavebands) were pre-assigned via software, and the imager was ready to work without any adjustments so that the user could concentrate on the patient.

After the patient number and lesion number was given to the system, the user interface guided the user to capture the dark and white references. The dark acquisition was performed by manually placing the imager to a light-blocking holder, seen in Figure 11. The imager was set to capture 40 frames, and the mean was used as a dark reference. The white reference procedure was similar—the imager was placed on a holder against white Teflon. The imager was set to capture with matching LED and wavelength settings, as it captures the HS images.

The user interface provided a preview video to target the image while the HS imager was placed on the patient’s skin. Three different LED light and waveband combinations were captured with a single process and one click to the capture button. With one click, the system captured six HS images. After capturing the HS image, the software calculated reflectance frames automatically for each wavelength and visualized those frame by frame in an animation view. The quality of the HS images was ensured visually from the animation by the users. At the end of the effortless workflow, the user could save or dismiss captured HS images with one click.

The user interface, the HS imager, and the chosen test setup (LEDs and correct wavebands) were pre-tested at the hospital with the users before the imaging for the clinical pre-test was started. The users were educated on the software and HS imager. The imager, dark and white reference targets and a computer with capturing software were mounted into a trolley so it could be stored in a secure place while unused (Figure 11a).

### 3.3. Materials and Methods

#### 3.3.1. Spectral Data and Pre-Processing

One HS image consists of three sets of frames, captured with thirty-three selected wavebands. The visible light (VIS) had two LEDs on, and the visible and near-infrared (VNIR) bands had seven LED lights on during the capturing process. The selected wavelengths and corresponding led lights are shown in Table 1. Each set of the thirty-three wavebands was captured three times, each time with one of the three different angles of light.

The HS images of the lesions had identification numbers. The dermatologist made the clinical diagnosis for each lesion before imaging, and the final diagnosis was confirmed with histopathological analysis of each lesion. The dermatologists hand-drew the ground truth images based on the histologically confirmed diagnoses.

The images were saved as raw images and pre-processed twice. We call these stages raw image pre-processing and machine learning pre-processing. The raw image processing pipeline is described in Table 2. First, the radiance images are processed into combined reflectance images. After that, the photometric stereo, described in detail in the next section, is calculated based on the hyperspectral reflectance images. The resulting dataset contains an albedo image and depth map for each measurement.

#### 3.3.2. Machine Learning Pre-Processing

Image quality reasons limited the number of the HS images from 14 BCCs to 10 and 8 IDs to 7. After the raw image pre-processing, each of the 17 selected HS image consisted of 33 albedo and 33 depth frames from the originally selected 33 wavebands.

In the first phase, we selected the most significant depth frame, which was the frame representing the wavelength of 575 nm in this test setup. The wavelength channel was selected based on the measured data’s robustness while ensuring that the channel is well within the range of visible light. Using only one depth frame, we could reduce the unnecessary depth dimensions from the data. After reducing the dimensions, the data were normalised between 0 and 1, and the possible infinity and non-numeral (NaN) values on the areas outside the imagers field of view were set to 0.

Images were vertically sliced from the middle of each lesion. The training pixels from the histologically confirmed lesions (250) and the healthy skin (100) were randomly selected from the left side of the HS image. The testing data were selected similarly from the right side of the image. The healthy skin pixels were selected by using a healthy skin mask, which was hand-drawn based on the ground truth images and ’RGB’ visualisations. The pixel selection is visualised in Figure 12. The ground truth labels of the selected pixels were selected accordingly.

The training and testing subsets from the HS image were rolling window views of the selected pixels. The size of one window was 30 times 30 pixels with 34 channels, and the step was set to 5. The training subset with its 6160 windowed HS-sub-cubes were balanced with Imbalanced learn library’s random over-sampling method [31]. After balancing, the data were augmented with vertical and horizontal flipping. The final size of the training data was 31,168. The validation data size was 1190. The test data had 5950 similarly windowed pixel HS-sub-cubes and their ground truth. The pixel-wise classification was performed using these randomly selected samples.

The classification maps shown in Section 3.4, were produced only for visualising the pixel-wise model’s potential and challenges of classifying and delineating whole HS images. For those classification maps, the whole HS images were pre-processed and windowed as described above. The accuracy metrics presented in this study (Table 3) are based on tests conducted with the pixel-wise analysis, using the above mentioned test data pixel-wise HS-sub-cubes.

#### 3.3.3. Method Validation

We validated the results by a leave-one-out approach to see if the pixel-wise slice half method causes biasing. The approach was conducted so that convolutional neural network (CNN) models were trained 17 times, omitting one lesion’s HS-sub-cubes (250 windowed lesion and 100 windowed healthy skin pixel HS-sub-cubes, size of 30×30×34) from the training material. Each model was tested separately. The test data were randomly selected from the pixels on the right side of the omitted image (250 windowed lesion HS-sub-cubes and 100 healthy skin windowed HS-sub-cubes).

As a control for this setup, each of the 17 lesions were classified with the original slice half model, trained with the 31,168 left-side HS-sub-cubes. Each of the image-specific pixel-wise HS-sub-cube results were calculated similarly to the leave-one-out approach by selecting similar windowed 250 lesion HS-sub-cubes and 100 healthy skin cubes from the right side of the lesion HS images.

The study was implemented with Scikit-Learn [32], Scikit-Image [31], SciPy [33] and Tensorflow [34] Python libraries. The computing was performed with a Linux GPU server, 1 × Tesla P100, ×86_64.

#### 3.3.4. Cnn Pixel-Wise Classifier

As the used data contain spatial and spectral domains and one depth map from the maps constructed from visible light for the classification, the natural choice for the classification is CNN [35]. With the convolutional neural network’s 3D and 2D layers, characterization of the data prior to the classifier is not needed, and the neural network is free to find connections invisible to human eye. In prior studies, the CNN has been deemed appropriate for tasks similar to the task in this study [3,11,15,36].

The limited availability of the labelled training data is one of the noted challenges related to the CNN classifier. Without a considerably large amount of training data, the models have a tendency to overfit [3]. Therefore, the selected classification approach was pixel-wise. The original 17 HS cubes were transformed into training data for the pixel-wise slice half model, consisting of 31,168 windowed 30×30×34 HS-sub-cubes after data augmentation, validation 1190 HS-sub-cubes. The test data were, respectively, a set of 5950 HS-sub-cubes.

We used a convolutional neural network (CNN) for the classification. Figure 13 visualises the structure of the network. The 3D convolutional layers were used to extract features from the windowed 30×30×33 HS albedo-sub-cubes. The construction of the 3D convolutional layers included the LeakyReLu activation function and max-pooling layers.

The 2D convolutional layers extracted the features from the windowed 30×30×1 HS depth map sub-cube. The 2D layers were constructed with LeakyReLu and max-pooling layers. The results were flattened, concatenated and used as an input for the hidden layers. As a result, the model provided the pixel-wise output classification and prediction confidences for three classes: healthy skin, intradermal nevus, and basal cell carcinoma.

The model was trained using Adam optimizer with default parameters and the categorical cross-entropy loss function.

### 3.4. Results

#### 3.4.1. Photometric Stereo and Albedo Spectra

To assess the quality of the photometric stereo transformation of the reflectance images, we take a look at three surface models: Surface model of a ball (Figure 14a), a Lego-brick (Figure 14b), and skin (Figure 14c,d). From them, we can see that the shape’s of the objects are visible from the surface models. In these cases, the interesting part is in the middle of the imaged area, and therefore there is very little error in the images. However, as the light direction matrix is defined for the image area’s centre, the error increases as the distance from the centre increases. Figure 14a–d are examples of cropped images, from the area that has a good quality. Outside the selected areas shown in the Figure 14a–d, the quality of the surface model was significantly weaker. These photometric stereo results were taken into account on selecting the training and test data by limiting the healthy skin area from the image border areas.

An example of albedo spectra with their deviation can be seen in Figure 15.

#### 3.4.2. Pixel-Wise Classification with Slice Half Model

The pixel-wise classification results of the slice half model in Table 3 show the precision, sensitivity and accuracy scores of the windowed HS-sub-cubes. The sensitivity is 0.81 with healthy skin and intradermal nevi and 0.76 with basal cell carcinomas. The average weighted precision was 0.81 and the accuracy over the whole testing dataset was 0.79. The results confirm that the model can distinguish the malignant and benign lesions at a pixel level. Since SICSURFIS HSI’s pixel resolution is approximately 24 μm× 24 μm, one classified pixel is smaller than one cell.

Confusion matrices (Figure 16) visualizes the pixel-wise classification results of the test data, shown in Table 3.

Figure 17, Figure 18 and Figure 19 are visualisations of the classified HS images. The images were conducted using the slice half approach model trained with 31,168 HS-sub-cubes. The classification method was pixel-wise, and these collages were conducted only for visual evaluations of the model’s capabilities and challenges. The HS images were pre-processed to HS-sub-cubes similarly to the training and test data. The classification prediction confidences and classification maps can be seen on these collages.

#### 3.4.3. Leave-One-Out Validation

This subsection compares the results obtained with a pixel-wise slice half model and 17 leave-one-out models. The results were obtained using subsets of the windowed (30×30×34) HS-sub-cube training data. For each lesion, the test data were selected from the right side of the lesions, which was not used to train any of the models. With leave-one-out validation, the test lesion pixel HS-sub-cubes were left out from the training subset cubes, and for the slice half model, the model was trained only using the HS-sub-cubes of pixels selected from the left side of the lesion.

Figure 20 visualises the pixel-wise classification accuracy comparison results for each lesion, which is the approach validation mentioned in Section 3.3.2. Y-axis represents the sliced data results, which model trained with 31,168 windowed HS-sub-cubes. The x-axis represents the results obtained with 17 leave-one-out models. The figure show which lesions pixel-wise classification accuracies correlate, and indicates which lesions had unique features.

The leave-one-out validation results show how the leave-one-out models were sensitive towards special features in the training data. Some of the individual lesion’s features and skin sub-types were unique. It affected the leave-one-out results decreasing it significantly. We can see from Table 4 that when the model is trained by leaving out windowed pixel HS-sub-cubes of a unique lesion, the model is unable to classify it correctly. It naturally has a strong descending effect on the leave-one-out average accuracy of all 17 models. For example, ID 17 (Table 4, ’RGB’ image in Appendix A) was the only lesion covered with hair. The model with no hairy pixels in the test data could reach 0.26 accuracy, while a slice half model with hairy pixels in the training data could reach 0.94 accuracy.

Eight lesions can be considered typical without any special features. When the training data contained pixel-wise HS-sub-cubes collected from those lesions, leaving one lesion out, the leave-one-out average accuracy those eight models was 0.72. The corresponding average accuracy of the pixel-wise slice half approach with the same lesions was 0.89. Nine lesions had unique features, of which two lesions were unique and difficult for both approaches. When leaving the pixel-wise HS-sub-cubes of one of these unique lesions out of the training data, the classification accuracy of these special lesions’ windowed pixel HS-sub-cubes decreased significantly. The average accuracy of those nine models was 0.28 in the leave-one-out approach. Therefore, the average accuracy of all of the 17 leave-one-out models is affected by the unique features explained in Table 4.

Table 4 and Figure 20 confirm that the results obtained with training data containing typical pixel window HS-sub-cubes from lesions with typical features among the training and test set, performed with significantly higher accuracy in the leave-one-out approach than the results obtained with unique test data. The average performance of the leave-one-out approach with typical data (0.72) validates the results of the slice half pixel-wise approach (0.89), indicating correlation and the model’s possibilities of generalization.

## 4. Discussion

The aim of this study was to introduce and demonstrate with a clinical pre-test a novel spectral imaging system designed for the complex skin surfaces. The pre-test demonstrated and enabled the system engineering, imaging, and analysis aspects of further system development toward a clinical application. The discussion is divided into six subsections, from technical and user-related topics (Section 4.1) to skin surface models (Section 4.2), annotation (Section 4.3), results (Section 4.5), approach validation (Section 4.6), bias conversation (Section 4.7) and finally to the notes for future research and the mentioned independent continuation studies (Section 4.8).

### 4.1. Technical and User-Related Issues

The SICRURFIS HSI had some issues related to the image quality or lack of healthy skin pixels with the smallest stray light protection cones. Those observations were technical, use-case related issues that can be solved in the future by changing the imaging strategy or by improving the device. With those user-related issues, some system engineering related topics were found.

The HS image capturing process was streamlined by only measuring wavebands relevant to the biophysical qualities of the skin and the illumination profiles of the used LEDs. However, the process still takes enough time, the minute movements of the medical professional using the device and the patient introduce a source for noise and inaccuracy. To solve this, image preview was implemented to the software’s graphical user interface and by applying spectral smoothing to the data. However, some motion of the operators hands is to be expected with the setup, and it may have contributed negatively to the image quality. In further research, a way to lower the capturing time and reduce the strength needed to operate the camera stable could be found by using new HSI technology. Interferometers can be manufactured as a micro electro-mechanical system (MEMS) using novel atomic layer decomposition techniques. One of these prototypes was presented by Trops et al. (2091). MEMS-based HS imagers can be significantly smaller and lighter with a faster frame rate. Besides the HS imager’s technical properties, the capturing time depends on the amount of the selected wavebands, so it could be further narrowed based on the suspected lesion type, making the image capturing faster.

Some of the images had quality issues which led to the decision to leave them out of the study. One of the problems was the sharpness of the images. The device required manual focusing, and some of the surfaces were challenging. For example, the nasal tip could be too small to the field of view, and the camera could not be set to the skin without passing some light from the sides of the nose. The unwanted light could be removed by covering the side areas with a hand, but it might make the manual focusing and staying on a place while capturing more difficult. Another focusing issue might be that the device is in sharp focus when the target is 6–6.5 cm away from the lens. Small and complex skin areas (e.g., nasal tip, ear) might come naturally closer than the required distance since the target might get inside the stray light protection cone. Similar quality issues have been raised in previous studies, e.g., in the study by Salmivuori et al. [10] uneven surfaces impaired the quality of the images.

The size of the stray light protection cone affected the usability of the images. Some of the images were captured with the smallest stray light protection cone. Those images had too small areas of healthy skin around the lesions. With the surface models, the best results were obtained with the stray light protection cone diameters of 4.9 and 5.8 cm. Additionally, the LED setup caused some reflections on the skin surface, which was difficult for our model to classify. For future studies, we would recommend capturing multiple pictures from the same patient, both images of lesions and separate images of only healthy skin. That would enable using the smallest stray light protection cones in the study because the healthy pixels could be selected from a different image. The reflectivity of the skin is one issue to be assessed in future research. It could be solved with linear polarizers placed in front of the LEDs.

According to the user feedback, the imaging system was easy to use and the technical issues found can be improved with system engineering. As seen in the results, the SICSURFIS HSI reached complex surfaces, which were mentioned to be excluded in previous studies (e.g., [14]). Eye corners, chin, shoulders, ears, neck, arms and other challenging skin areas were reachable, and the pixel-wise classification results were promising.

### 4.2. About the Skin Surface Models

In the calculated surface models, the accuracy deteriorates as a function of the distance from the image center. In the border areas of the images, the effect is visible from the visualisations in Figure 14. The deterioration is an inherent feature of the used Frankot–Chellappa algorithm, as the light direction is defined only for the center of the imaging area. The poor accuracy may negatively affect the classification results, and it would be beneficial to study how to use individual light direction matrices for each point.

In our measurement setup, there are many moving pieces, and therefore the process of determining the individual light direction matrices was deemed inaccurate. The surface models are affected also by the penetration depth of the used light.

In this study, we decided to use wavelength from visible light (approx. 575 nm) as the basis of the surface model, and used it along the 33 albedo channels. The CNN’s 2D layer extracted the features from this wavelength. The results are promising, but there is a need for further research on the benefits of the surface model’s usage in HS image pixel classification. It could be beneficial to study the surface models of more penetrating wavelengths. If the density of the skin is significantly different between cancerous and healthy areas, the difference could be made visible with surface models formed with infrared lights.

### 4.3. The Annotation and Pixel Selection

The dermatologist drew the ground truth of the lesions and healthy skin. The poor signal-to-noise ratio of some of the RGB constructions might have affected on drawing of the edges of the annotated lesions. Therefore the rules of how we selected the pixels could be improved. We did not consider that there might be pixels close to the lesion edges that are incorrectly classified as healthy skin or lesion on the ground truth.

For future machine learning preprocessing, it might be necessary to select the healthy skin pixels as far as possible from the annotated lesion and leave some margin to the edges of the lesion. This challenge could also benefit from the purely healthy skin HS images captured from the same patients.

### 4.4. Notes from the Pixel-Wise Analysis

According to Ahmad et al., the lack of labelled HS data is a major issue since labelling is time-consuming and expensive due to human experts and investigation. In this pre-study, the HS images were annotated by dermatologists, and the ground truth was confirmed with dermatohistopathological investigation, which is a strength.

Since the CNN models have a tendency to overfit when the amount of the HS training data is too small, we selected a pixel-wise classification approach with CNN, which is a common strategy with HS data and deep neural networks [3]. The training data of the CNN was constructed by slicing the HS images into two images per HS image and using the left side as a source for the selected training data pixels and the right side as a source for the testing data pixels. Instead of training and testing the model with only 17 HS images, we could utilise the spatial, spectral and skin surface domains pixel-wise with spectral pixel sub-cubes. After the data augmentation, the training set consisted of 31,168 windowed pixels (30×30×34). Before the augmentation, from each lesion HS image, there were 250 lesion pixels and 100 healthy skin pixels, which were randomly selected as the middle points of those windowed HS-sub-cubes. The test data were collected similarly from the right sides of the lesion HS images. This way, we achieved satisfactory amount of data points for both training and testing.

The CNN seems to perform well in classifying the malignant BCC (nodular and superficial), benign ID and healthy skin lesion pixels. The nodular BCC’s confidence map (Figure 17) shows that the model correctly does not find any ID pixels from the image data pixels. It draws pixel-by-pixel quite accurately the shape of the elevated lesion. The prediction confidence map shows some BCC pixels in the right upper corner, which are, according to dermatologists, false classification results. Nodular small (<1 cm) BCC are usually sharply demarcated. The width of the lesion was 5 mm. The lesion is located on the upper eyelid. The skin around the lesion has multiple colours in the image, and there are some wrinkles and light reflections. The model has classified some of the healthy skin pixels with reflections and areas with healthy skin pixels containing skin colour changes from dark to light reflection as BCC pixels.

The second example collage (Figure 18) is a 12 mm nodular and superficial BCC on the left upper arm. According to the dermatologist, the superficial parts are usually in the periphery of the lesion and can have indistinct borders. The CNN draws a BCC lesion pixel-wise with indistinct edges and some satellite lesions around it. There is a possibility that satellite lesions can occur in the periphery of a superficial BCC that are impossible to detect by the human eye. These classified lesion pixels in this study might indicate that with HS imaging and machine learning methods, it could be possible to provide information that guides the dermatologists to delineate and remove a lesion more accurately. SICSURFIS HSI’s pixel resolution is approximately 24 μm× 24 μm, one classified pixel is smaller than one cell, which can be seen as an advance for applications requiring high accuracy.

Third collage (Figure 19) is an example of a benign intradermal nevus pixel classification results. The lesion is located on the right side of the back of the patient. The lesion is delineated and classified accurately, but there are redness and reflections in the pixels surrounding of the lesion that are miss-classified as nevus pixels. Intradermal nevi are sharply demarcated, and no satellite lesions should be seen. Some of the pixel level challenges relate to the healthy skin pixels. In the future, larger samples of HS images of healthy skin could improve the classification results of healthy skin.

### 4.5. Sensitivity and Precision

The numerical results of the pixel-wise slice half model’s test data (Table 3) show relatively good weighted sensitivity (0.79) and precision (0.81) for the model.

In our study, the sensitivity is mainly impaired by inaccurate delineation of the lesions, which again seem to be caused mostly by light reflections and uneven colour of the healthy skin in the images. The sensitivity is also decreased because the sensitivity calculations compare the diagnosis separately for each pixel of a lesion, not by a voting method with one diagnosis per lesion only. Our test data pixels were selected randomly from the right side of the HS images lesion and healthy skin areas, and since the CNN is a three class pixel-classifier, the results of those pixels may contain two types of lesion or healthy skin pixels.

These sensitivity and precision results are not directly comparable with previous studies (e.g., [36]), which uses the majority voting method on the annotated lesion areas. In the majority voting approach, the whole area of the annotated lesion is pixel-wise classified based on the majority of the pixels, so there cannot be, for example, two types of lesion pixels on the same lesion area.

As an important note, the used data contain an inherit selection bias which in some cases increases and in some cases decreases the accuracy. The slice half method also cause bias by increasing the the obtained accuracy. Therefore, the results are promising, but further research is needed. In the further studies, the results could be addressed also with a majority voting method, which would enable more discussion with previous research.

### 4.6. Approach Validation

Besides the promising results, this kind of clinical pre-study has limitations and concerns related to approaches and methods. The pixel-wise classification approach, using only one HS cube as a source for training, validation and testing is a common approach in HS image analysis and classification studies [3] (examples in [1,37,38]).

One of the possible downsides of the pixel-wise slice half approach is that after the training, the neural network has been presented with data very similar to the test data, which might bias the results. For observing the possible bias, the pixel-wise slice half classification results were validated with a pixel-wise leave-one-out approach.

As a result, the leave-one-out validation metrics were strongly decreased with unique lesion features (see Table 4, Figure 20 and Appendix A). Only the results obtained with models trained with typical data should be considered a validation; the leave-one-out models trained with data containing windowed pixels (HS-sub-cubes) from the eight typical lesions reached 0.72 average accuracy, varying between 0.83 and 0.64. The average accuracy of the same eight typical lesion’s pixel-wise HS-sub-cubes with the model trained with slice-half approach and 31,168 HS-sub-cubes was 0.89, varying between 0.96 and 0.79. This finding supports the potential results of SICSURFIS HSI’s capability to classify, differ, and delineate the BCC, ID, and healthy skin pixels, and validates the approach.

The decreasing effect in the leave-one-out results is visible in Figure 20 and Table 4. We can see that the accuracy of the pixel-wise test data (HS-sub-cubes) collected from lesions 1, 4, 7, 8, 9, 10, 14 and 15 were correlating, which indicates that those lesions had some typical pixel-level features that enable both approaches models to generalise. The effect can be seen with nine lesions; the pixel-wise test data collected from lesions 2, 3, 5, 6, 11, 12, 13, 16 and 17 had some special features, making the lesions unique in the training set. Therefore, nine of the seventeen leave-one-out models could not generalise well. These models could not classify a pixel that was vastly different from the training data set. For example, there was only one hairy lesion, only one red-brown lesion or a single lesion with a scab. Those features in the collected pixel-wise HS-cub-cubes could be a reason for a model not to be able to classify pixels containing those features correctly. Two of the lesions, 6 and 16, were difficult and unique, which can be seen in the pixel-wise accuracy of both approaches.

The approach validation shows that the results obtained with eight typical lesions and both approaches are promising. The level of bias is acceptable for a study that has the purpose of the first demonstration of a prototype imager and points out the future improvement needs. The slice half results are higher, and the leave-one-results lower, but the model’s capability to generalise with unbiased data can be seen, but further studies and more data are still needed.

For closer investigation of the results and validation, the Appendix A presents the RGB-reconstructions, histopathologically confirmed ground truth, accuracy results of both tests, and clinical details (lesion type, size, location) and possible mentions of special features. Based on the special features that are visible in RGB reconstructions, the lesions were divided into three classes, grey (typical), red (unique) and purple (unique and difficult). The colors and image IDs match with the results shown in Figure 20 and Table 4.

### 4.7. Selection Bias and Other Data-Based Biases and Their Effect to the Results

Selection bias is “*A systematic error that results in differences between a study population and a target population; selection bias primarily affects the external validity of the results of a study*” [39]. In this study, the selection bias concerns the study population and study lesions. The variability of the real target population (humans all over the world) is not fully covered and the variability of different sub-types of lesions (i.e., hairy, growth style, different pigment variations, wounds, blood, thin or thick skin etc.) has the same low coverage.

When developing an application to clinical imaging, the selection bias is probably the most important reason for a system with high accuracy on pre-tests to be unreliable in clinical use [39]. This effect has been noticed for example in machine learning algorithms utilised with radiology images. The recent study points out the selection bias as the possible reason, why the results at the hospital with real patients are less expressive [39]. This research, and the SICSURFIS HSI, as it is, is a long way from a clinical application. According to Yu and Eng, the selection bias has been largely unaddressed in the medical imaging machine learning literature. Bias in the collected data causes distortion among the results. For example, the classification accuracy can increase or decrease, but the level of bias can and should be examined and controlled [39,40,41]. Untrained unique features decreases the accuracy and features that are covered in the training data enables higher generalisation and accuracy.

As shown in this study, the bias is difficult to avoid. It can be caused by a method or data. The development of a real-life medical application is expensive. There are plenty of sources of bias that affect any results: patient skin tone (native Asian, north European, African vs south European), clinical practices, patient age distribution, users of the imagers and so on. If the inherent bias is not caused by a method, it is caused by a data set. The level of bias is important to estimate, and further research should address decreasing it. Deep learning algorithms, such as CNN, is detecting and distinguishing features automatically, giving them weights. Basically, it is a black box, which does not explain why the diagnosis is made [39]. In this study, the selection bias is obvious; the number of lesions were relatively low, and the selected lesions had strong variation. Those, that had unique features failed on the leave-one-analysis, showing, that in order to develop a more robust system and a real application, it would require a enormous amount of data, captured in many countries, with a wide diverse among the patients and remarkable amounts of lesions.

Traditional image classification is different than pixel-classification mainly due to the spectral domain, and the fact that each pixel is classified one-by one. Previous image classification studies show, that in image classification, it is possible to train the model and classify sub-images with no sensible, visually interpret information, and depending on the sub-images size, to obtain even good accuracy levels. The reason of this is, that the model might be able to to detect and weight features also from the background; use other visual aspects the human eye does not see features [40]. The size of the sub window seemed to correlate with the accuracy. With all tested known standard data sets and regardless of the level of bias factors, the small (20×20) subwindows reached lower accuracy than the larger ones (200×200) [40].

In our case, we used hyperspectral data and a pixel-wise classification approach. Each sub-cube contained albedo and depth channels, providing a spectrum for each pixel. Most of the training data were augmented, e.g., flipped, which differs the selected 30×30×34 sub-cubes from the originally selected training pixels. The test pixels were not flipped. The size of the original training set was 6160 samples and after the augmentation, the size was 31,168. Therefore, we utilised a small-sized samples from each HS cube to control the possible data-based bias levels, and we gave pixel-wise information for CNN to extract with 3D and 2D layers. This causes bias, since the source of test and training pixels is the same data set (HS cube from a lesion), but as shown in the above mentioned previous studies, the small size of the used sub-cubes was chosen to keep the level low.

The sources for bias are various. For example if several data sets are collected in a controlled environment and same session, all of these HS images might contain characteristics that are caused by the image acquisition. So far, the machine learning model might classify them based on the session, not by the features seen in images [40]. In our data, this can be one of the bias reasons besides the selection bias; a subtle changes in the lighting conditions (a stray light passes the cones) or even the temperature of the sensor at the time of imaging might have a result in artifacts that might lead to differences that are to be detected by machine learning models [40]. As an example, the stray light increases the illumination when imaging a lesion, and a global feature, such as pixel intensity statistics, can be used as a decision feature inside the CNN. This phenomena was controlled with the protection cones, but some of the surfaces were challenging and the data might contain individual images with features caused by the stray light. An other example is that if the imager is in use for several hours, the sensor temperature increases, and it can cause some features in the collected data; the dark-correction is performed for each HS cube. Dark reference images captured right before capturing the lesions, with similar exposure time, so the correction includes the current effect of sensor temperature. This might cause difference among the captured data, some of the HS images might have characteristics from the acquiring process, which influences to the results. One of the benefits of the randomly selected pixels and sub-cubes is, that if the sensor has dead pixels, that exists in every frame, this kind of feature is not automatically presented to the machine learning model.

As shown by this discussion, the perceived worth of this study might be diminished by the bias. However, the reason for this clinical pre-study was to introduce and pre-test a new imaging system. Without the first models and results, it would have been impossible to evaluate the technical aspects; how the imager reaches the complex surfaces, can those images be classified by the CNN. It was important to test, how the imager’s special LED module, stray light protection and optical components affect on the image quality and to the results, and finally evaluate the clinical procedures. Based on this study, for example, the smallest protection cone has been rejected from the following data gathering steps and the stray light blocking in the challenging areas is improved with extra protection. One could say, that the accuracy of the leave-one-out approach is the real performance, but it is not the whole truth. Another opinion is, that due to the unseen features, the leave-one-out results reflect the selection bias on the collected data.

The main point is not, what interpretation of the approaches and results is the right or how much and what kind of bias the collected data contains. The outcome is more in the potential, that the SICSURFIS HSI has, and in the results that can be obtained in the future via improvements. The results of the both approaches indicates that the proposed models would be practical with a larger amount of training data, which contains no unique lesions, and so far the selection bias could be minimised. The hypothesis for future studies is that the accuracy will increase when the collected data enables the models to generalise better. The accuracy difference between the leave-one-out method and the slice half method is expected to decrease as the data amount increases, and the level of selection bias should be taken into account in the further studies.

### 4.8. Future Research

In the following studies, we will increase the amount and variability of the data, improve the technical and user-related issues and capture more lesion types for developing a more effective CNN classifier to get one step closer to an optical biopsy. Current results are promising, but the need for further studies is obvious.

## 5. Conclusions

This article aimed to introduce and demonstrate a Fabry–Perot interferometer-based hyperspectral imaging system for complex surfaces. This study was the first step in a three-phase pilot study demonstrating the possibilities of using a new system as a first prototype for future real-life applications. This section concludes our findings in technical aspects (Section 5.1), methodological approaches (Section 5.2) and results, and future research (Section 5.3).

### 5.1. Technical Aspects

As the described device is a prototype, there are still some issues to fix before achieving results that promote the devices use in clinical use. There is some variation in the focus between the spectral channels. Some of the frames might be slightly more out of focus than others. Bigger light protection cones provided better quality data than the smaller ones. There might not be enough healthy skin around the lesions with the small cones. The LED setup caused some reflections to the skin areas, which were difficult to handle for the convolutional neural network model. The depth maps were calculated for each wavelength, and one of the noted issues was the quality of the 3D models. The best quality was in the middle of the image. The quality deteriorated towards the edges of the frames. It is possible to improve the depth of the focus in the whole wavelength range by replacing the used commercial S- and C-mount lenses with a dedicated custom-designed optical system. The specular reflections can be reduced by using linear polarizers in front of the LEDs and the imaging optics.

### 5.2. Classification Results and Method Validation

Based on the pre-test, the HS imager and machine learning system accurately differentiate malignant BCC from benign ID and healthy skin, achieving a weighted sensitivity of 0.79 and weighted precision of 0.81. The classification report reveals the results of the entire test data pixel-by-pixel classification. The results are not directly comparable with previous studies using voting methods for classifying pixel-wise the lesion types.

We used a pixel-wise classification approach with CNN classifier. The approach is common with HS images, due to the lack of labelled HS data. The number of HS cubes used in this study was 17, which is generally 16 HS images more than in a typical HS image classification method development experiments; it is typical to use the pixels from one standard data set, and divide them to training, validation and test portions [3]. In our approach, we sliced each of the lesion HS images and selected the training data from the left sides of the lesions. The training data consisted of 31,168 windowed pixels (HS-sub-cubes). Our test data consisted of similarly collected windowed pixels (HS-sub-cubes), selected randomly from the right side of the HS images lesion and healthy skin areas. The bright side of the pixel-wise approach is the large amount of the training data since it consists of sub samples. The CNN was not overfitted due to lack of training data, but the approach might bias the results. Therefore, the results were validated with a pixel-wise leave-one-out approach. Eight of the lesions consisted of pixel windows with typical features, providing relatively well-generalising models and correlating accuracy results with the sliced half approach. The rest of the leave-one-out models performed poorly due to special features in those lesions. Therefore, the average accuracy results of all of the 17 models in the validation were lower. For example, a model trained without hairy pixels could not accurately classify pixels collected from a hairy lesion image. These phenomena were seen with nine lesions and models that used the pixels collected from those lesions as test data, representing those unique features.

For the comparison of validation to the slice half approach, only the results of the eight typical lesions should be taken into account. The average accuracy of those results was 0.72, whereas the similar average accuracy with the slice half approach was 0.89. These results are shown in Table 4 and Figure 20. The validation confirms that this 3D approach has promising results, but there is a need for further studies with a larger amount of data with no unique images. The bias caused by the approach is seen in the slice half method, but the capability of models to generalise was proven with the validation approach. The accuracy results without bias might be lower than the achieved, but the results of this study might indicate that with HS imaging and machine learning methods, it could be possible to provide information that guides dermatologists to delineate and thereby remove a lesion more accurately. On the other hand, the number of captured lesions was small, containing unique lesions, impacting the results’ generalisation.

### 5.3. Conclusions for the Future

This was a development starting point and a demonstration of a novel SICSURFIS HSI system, which seems to have potential, but the need for further studies is obvious. With its results and notes, the pre-test pointed out benefits and findings for many future improvement aspects.

COVID-19 had a decreasing effect on the data gathering phase’s patient recruitment. As the system uses unique, specifically selected LEDs and wavelength bands, most of the previously gathered HSI data are inappropriate for training the machine learning algorithms used in the research. Therefore, data gathering with a similar device should continue in the future.

As the dataset size increases, the leave-one-out cross-validation method should be adopted. The accuracy difference between the leave-one-out method and the slice half method is expected to decrease as the data amount increases. There is no risk for cross-contamination between training and testing data in the leave-one-out method.

Before the clinical use, all of these issues should be inspected. Another technical step toward the future might be the MEMS FPI, which could reduce the size and weight of the handheld device. There is also a need for developing the system to be a real-time solution. It might mean the FPGA computation and a screen on top of the device for the doctors to see the results immediately.

## Figures and Tables

**Figure 1 sensors-22-03420-f001:**
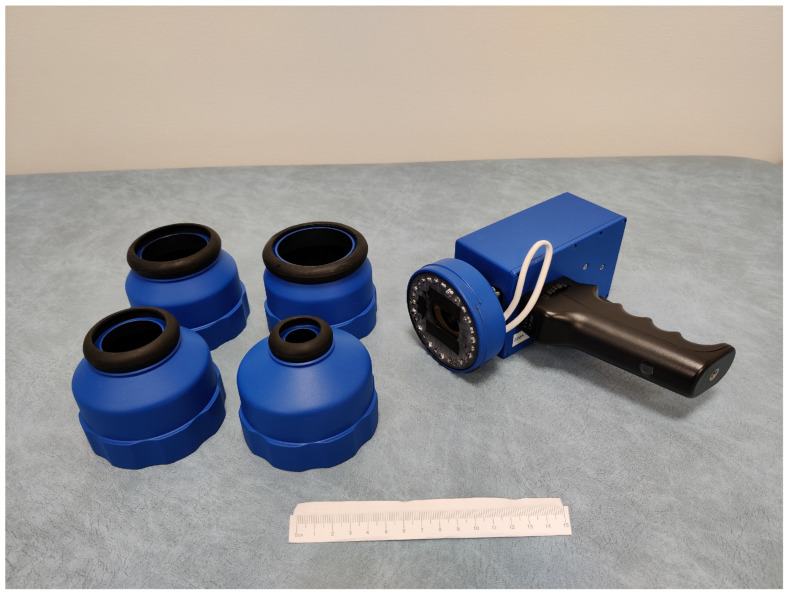
SICSURFIS spectral imager with integrated LED module and stray light protection cones.

**Figure 2 sensors-22-03420-f002:**
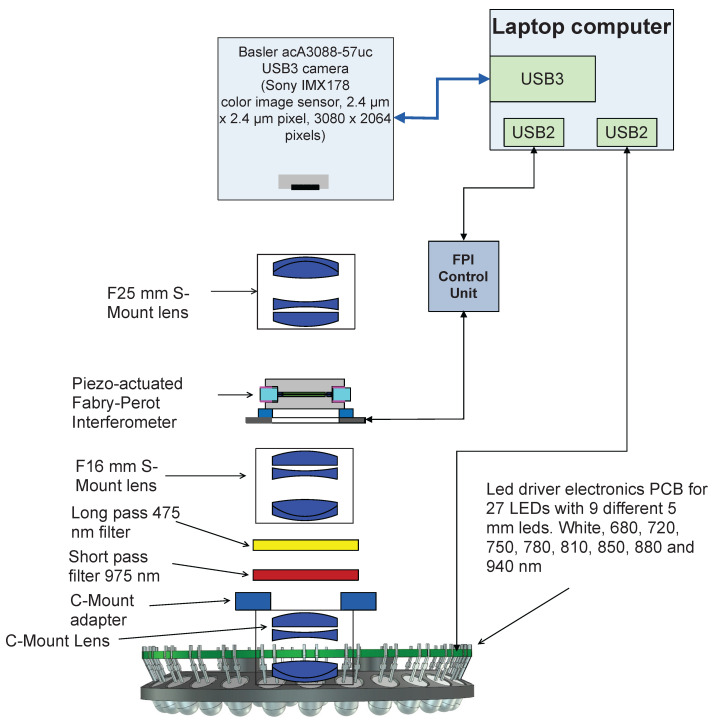
Block diagram of the SICSURFIS spectral imager.

**Figure 3 sensors-22-03420-f003:**
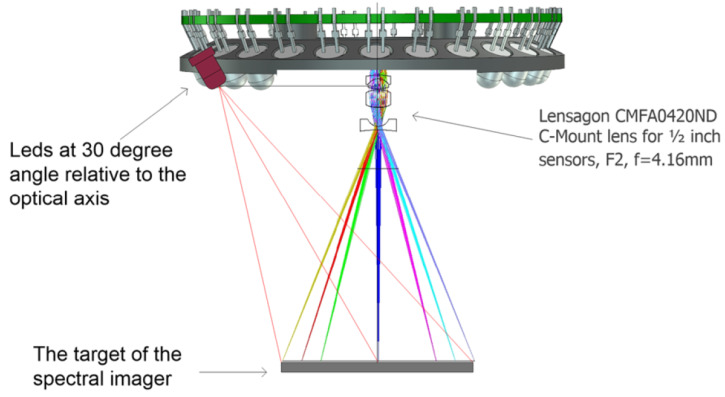
Three sets of nine LEDs are at a 30-degree angle relative to the system’s optical axis. These three same wavelength LEDs sets are located at 120-degree intervals on the Led PCB.

**Figure 4 sensors-22-03420-f004:**
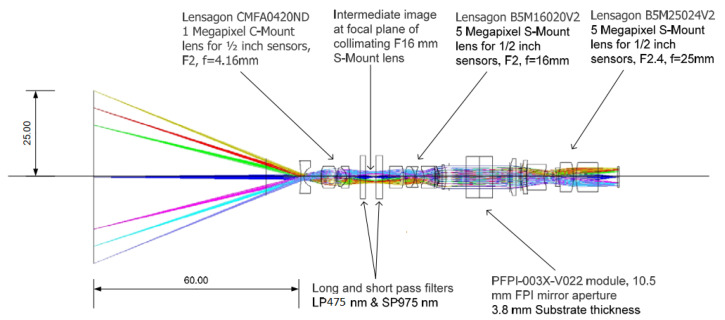
Optical concept of the imaging system of the SICSURFIS Spectral Imager.

**Figure 5 sensors-22-03420-f005:**
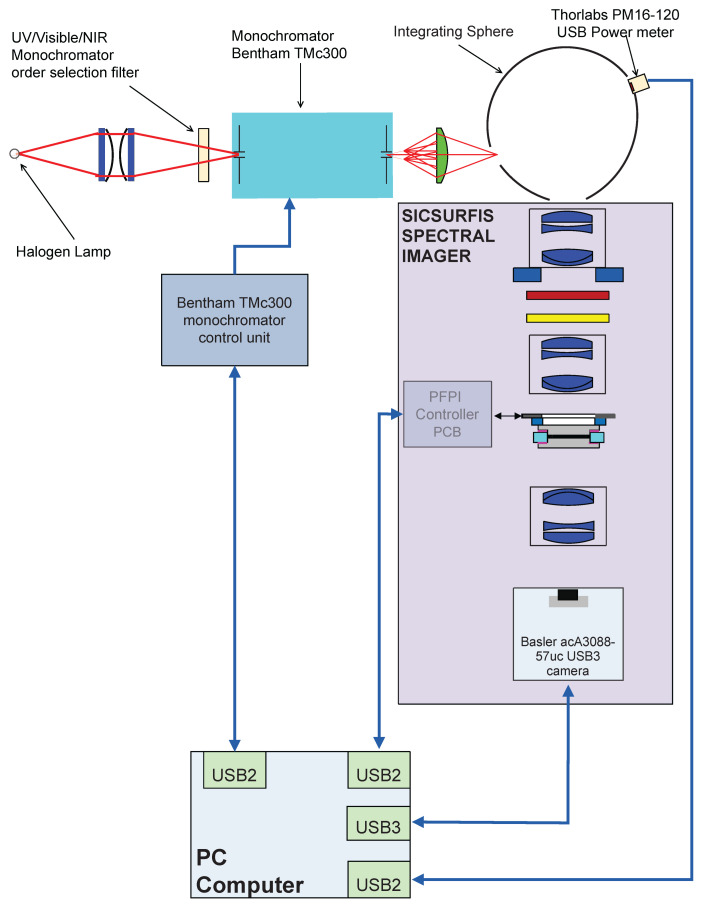
The SICSURFIS hyperspectral imager calibration measurement setup.

**Figure 6 sensors-22-03420-f006:**
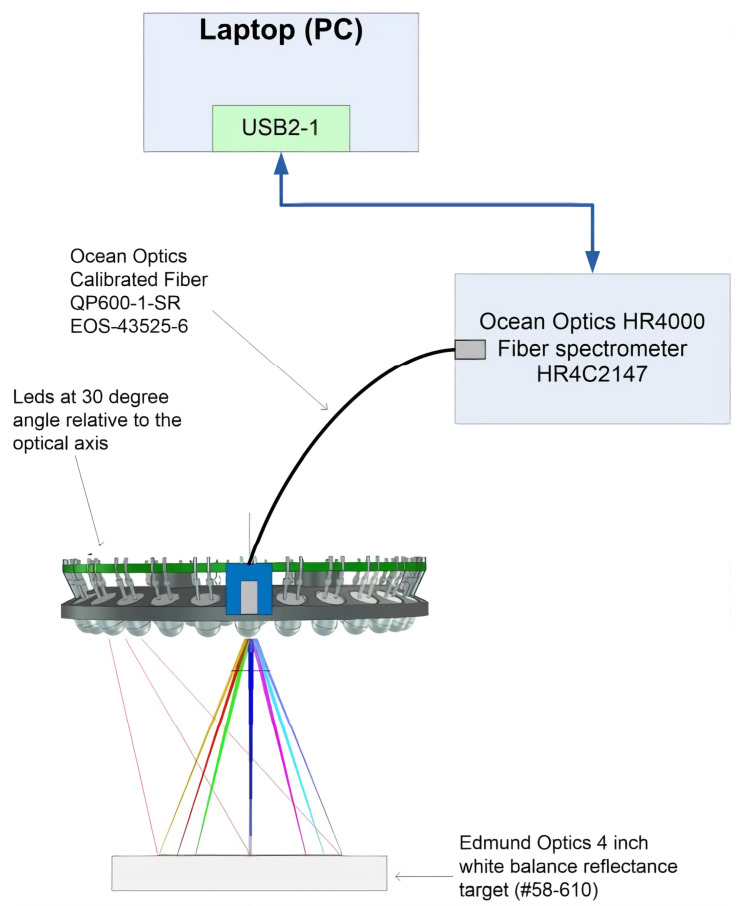
The setup was used in the spectral radiance measurements of LEDs and LED groups using the Ocean Optics Calibrated Fiber QP600-1-SR EOS-43525-6 and HR4000 spectrometer. Edmund Optics 4 inch white balance reflectance target (#58-610) was used in the measurements.

**Figure 7 sensors-22-03420-f007:**
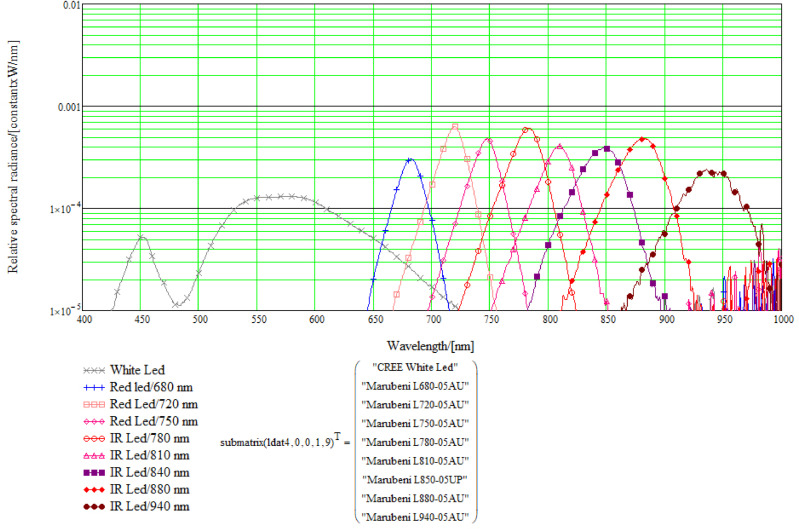
Results of the single LED spectral radiance measurements with the setup shown in Figure 6. The ma38 mA DC LED current was used for each LED. The distance of the target in this measurement was 50 mm.

**Figure 8 sensors-22-03420-f008:**
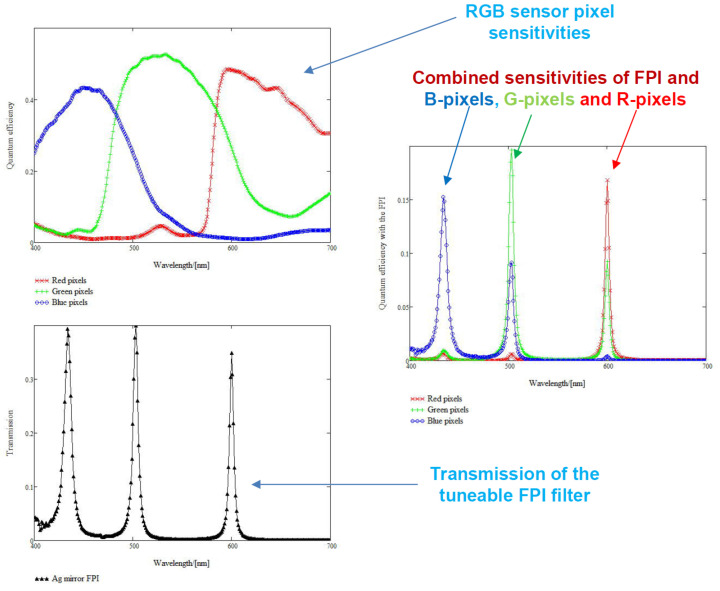
Quantum efficiencies of RGB CMOS image sensor (upper left) red, green and blue pixels, Transmission of metallic mirror FPI (lower left) and combined sensitivities of the FPI and red, green and blue pixels (upper right).

**Figure 9 sensors-22-03420-f009:**
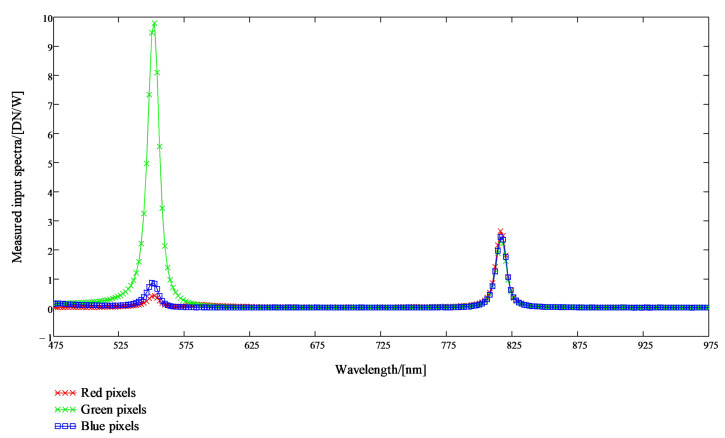
Results of the SICSURFIS HSI monochromator calibration. The scaled R-, G- and B-pixel sensitivity signals are plotted in units DN/(W/nm). There are two peak wavelengths for the selected PFPI drive voltage, 548.2 nm (on the left) and 812.2 nm (on the right).

**Figure 10 sensors-22-03420-f010:**
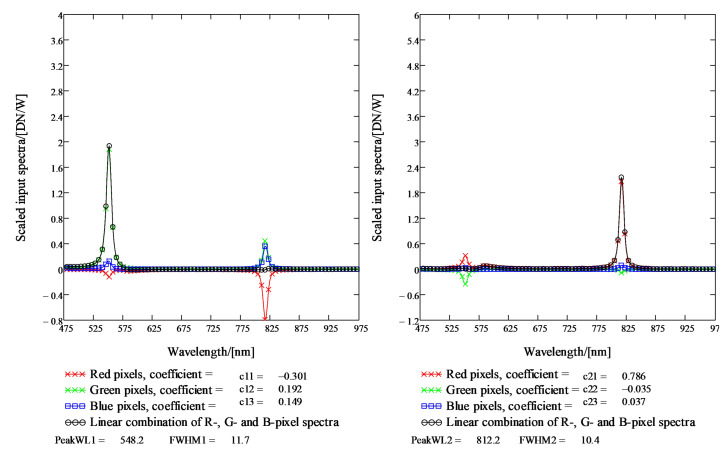
Results of the SICSURFIS HSI monochromator calibration. The linear combination of the scaled pixel sensitivity functions for the peak wavelengths 1, 2 and 3. For the selected PFPI drive voltage, there are two peak wavelengths, 548.2 nm (on the left) and 812.2 nm (at the center).

**Figure 11 sensors-22-03420-f011:**
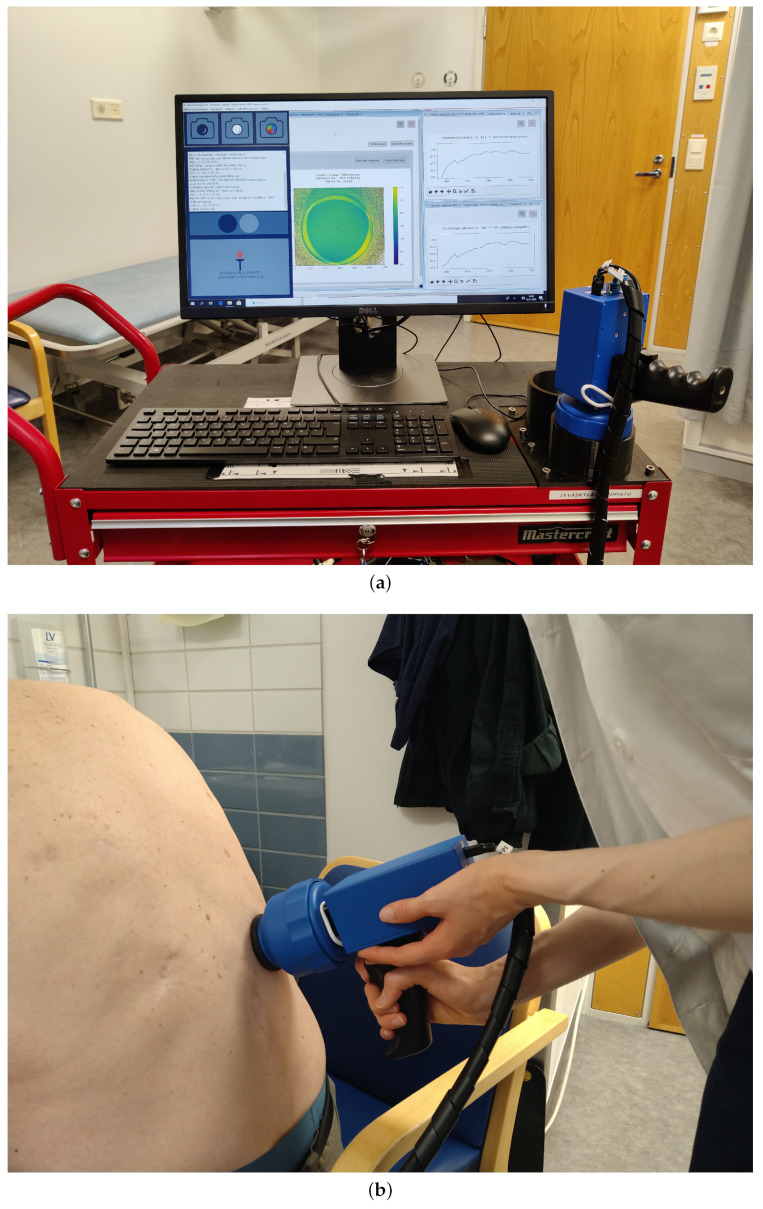
SICSURFIS imaging system. A handheld image sensor and a small computer for data collection are attached to a trolley that can be moved around hospital wards. Subfigure (**a**): SICSURFIS imaging system. Subfigure (**b**): SICSURFIS HSI in clinical use.

**Figure 12 sensors-22-03420-f012:**
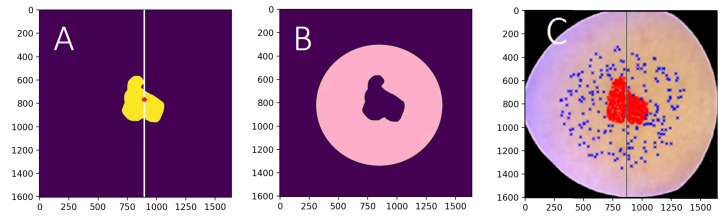
(**A**) The ground truth image, left side: training data, right side: test data. (**B**) The healthy skin mask is visualised with pink colour. (**C**) The visualisation of the selected training and testing data points. The selected lesion pixels are visualised in red, and blue represents the selected healthy skin pixels. The size of the HS image (**A**,**B**) was 1605 × 1640 px; the size of the divided training and testing portions (**C**) varies, depending on the lesion’s location.

**Figure 13 sensors-22-03420-f013:**
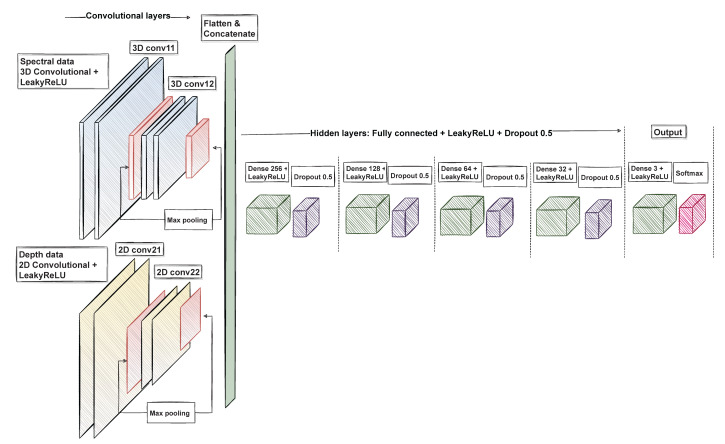
The visualisation of the used convolutional neural network. We used 3D layers with spectral data and 2D layers with depth data. The outputs were concatenated, flattened and used as an input for the hidden layers. The result was a three-class classifier for spectral data with depth information.

**Figure 14 sensors-22-03420-f014:**
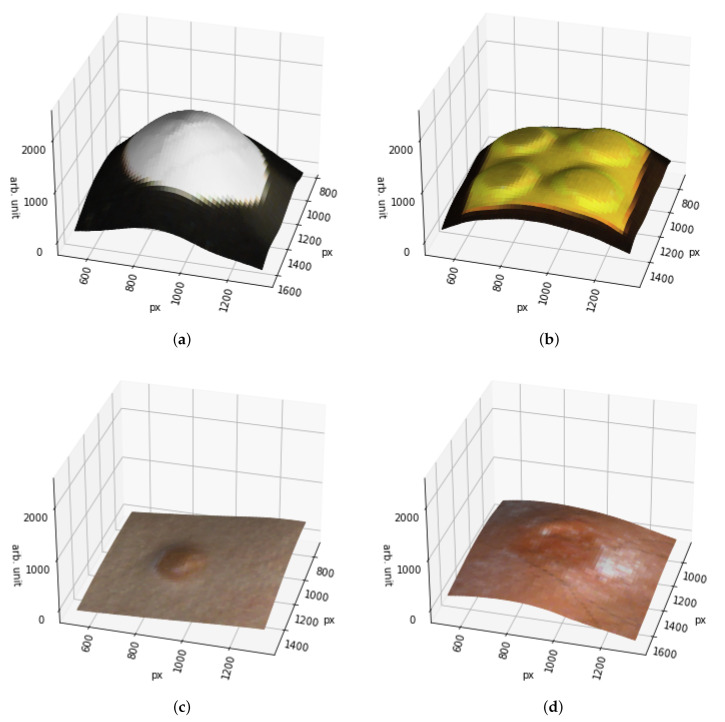
Surface models of (**a**) Small styrofoam ball, (**b**) Two times two yellow Lego-brick, (**c**) Intradermal nevus on human skin and (**d**) Human skin with nodular basal cell carcinoma.

**Figure 15 sensors-22-03420-f015:**
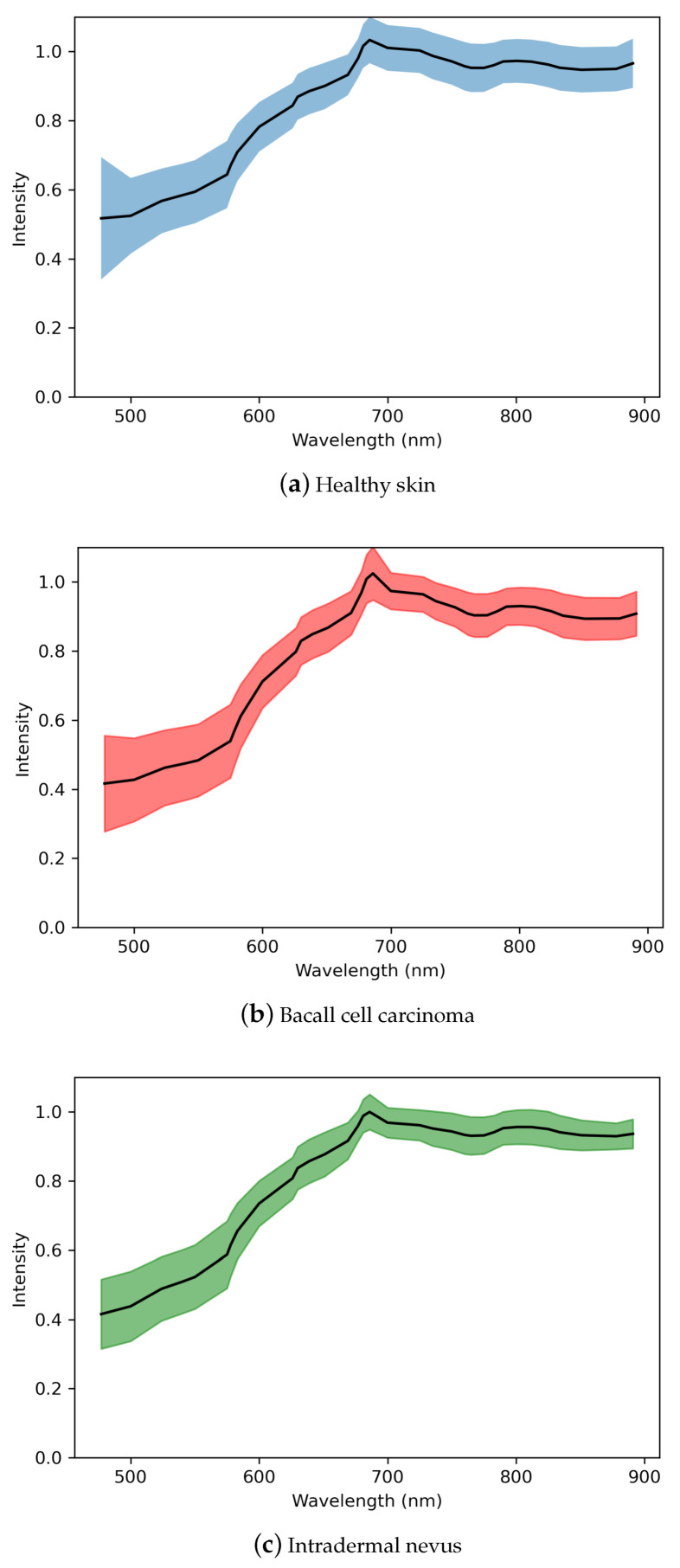
Sub figures (**a**–**c**) represent the albedo spectra of healthy skin (**a**), BCC (**b**) and ID (**c**). The minimum and maximum standard deviations are marked with coloured areas, and the mean albedo spectra are drawn with a black line.

**Figure 16 sensors-22-03420-f016:**
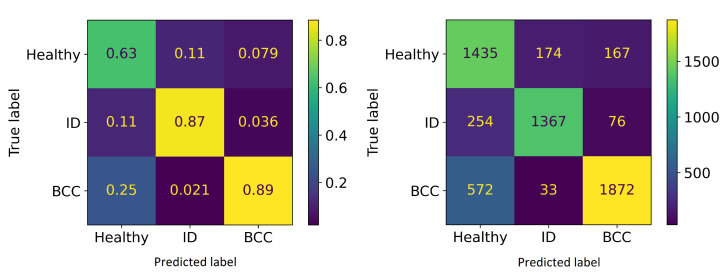
Confusion matrices. **Left**: Normalised predictions, **Right**: predictions.

**Figure 17 sensors-22-03420-f017:**
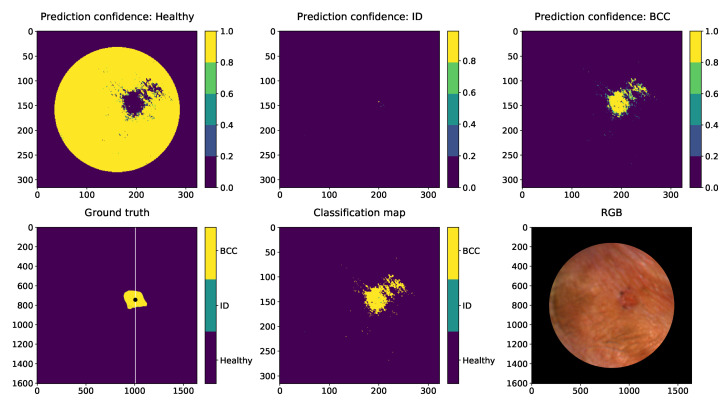
Nodular BCC visual pixel spectra classification example. First row: Healthy skin prediction confidence, prediction confidence of the ID and the prediction confidence of the BCC. Second row: The ground truth. The white line in the ground truth image shows the slicing position. The training data were collected as windowed (30×30×34) HS-sub-cubes. The middle pixels of those HS-sub-cubes (250 lesion and 100 healthy skin pixels) were randomly selected from the left sides of the lesions. We can see from the classification map (middle) that the nodular lesion is delineated relatively well, but some miss-classified pixels can be seen above the right upper corner of the lesion. The “RGB” illustration of the lesion is on the right. The lesion was captured from the left corner of the eye. The unit of measures is smaller with prediction maps since the HS images were windowed with 5 pixels step.

**Figure 18 sensors-22-03420-f018:**
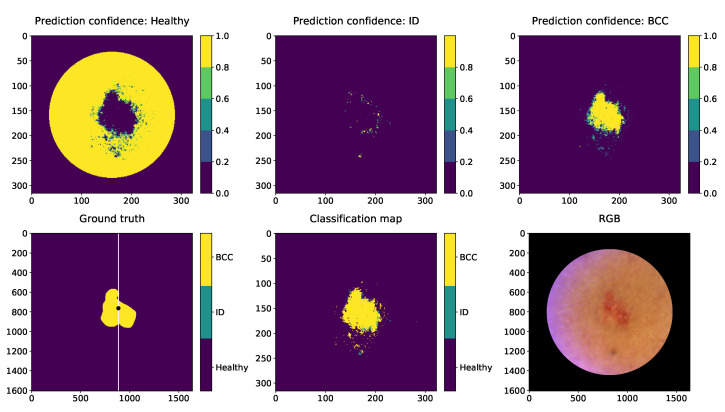
Nodular and superficial BCC visual pixel spectra classification example. First row: Healthy skin prediction confidence, prediction confidence of the ID and the prediction confidence of the BCC. Second row: The ground truth. The white line in the ground truth image shows the slicing position. The training data were collected as windowed (30×30×34) HS-sub-cubes. The middle pixels of those HS-sub-cubes (250 lesion and 100 healthy skin pixels) were randomly selected from the left sides of the lesions. The middle image is the classification map of this BCC HS image. “RGB” illustration of the lesion is on the right. The lesion was captured from the left upper arm. The unit of measures is smaller with prediction maps since the HS images were windowed with 5 pixels step.

**Figure 19 sensors-22-03420-f019:**
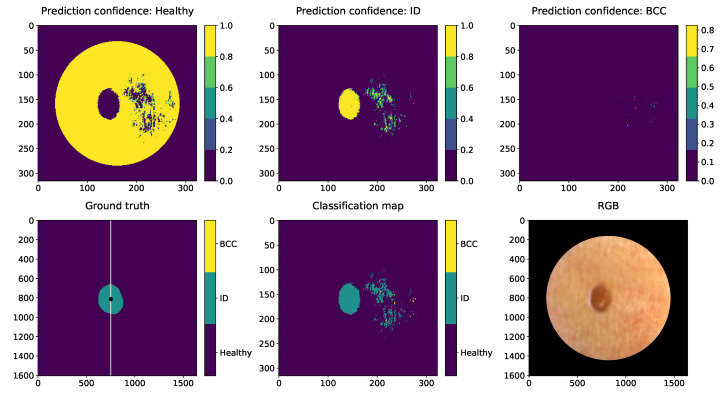
IN visual classification example shows that benign ID lesion pixels can be classified and the lesions can delineate efficiently. The classification map and ID prediction conference visualization show that some of the reflections or different shades of redness on the healthy skin might have confused the classification result. The lesion was captured from the right side of the patient’s back. The unit of measures is smaller with prediction maps since the HS images were windowed with 5 pixels step.

**Figure 20 sensors-22-03420-f020:**
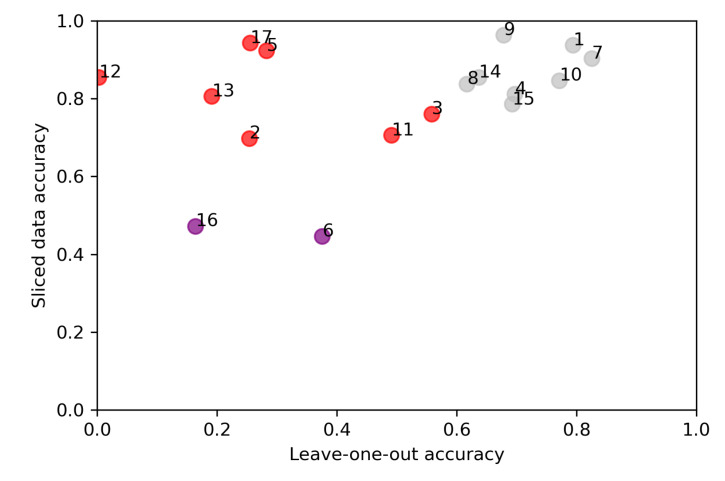
Comparison of pixel-wise accuracy results. Y-axis represents the results obtained with model trained and tested with sliced data. X-axis visualises the classification accuracy of leave-one-out models. The details of every HS image with RGB reconstructions can be seen in Appendix A. Grey, red and purple colours indicate the special features and the models’ capability of classifying typical (grey), unique (red), difficult and unique (purple) lesions in this test set.

**Table 1 sensors-22-03420-t001:** Selected wavelengths and corresponding LED light wavelengths.

**VIS**	477, 500, 524, 540, 550, 575, 578, 582, 600, 626, 630, 639, 651, 669, 677, 681, 686
**VIS LED**	white, 680
**VNIR**	700, 725, 735, 750, 760, 765, 775, 783, 790, 801, 812, 825, 834, 851, 878, 891
**VNIR LEDS**	720, 750, 780, 810, 850, 880, 940

**Table 2 sensors-22-03420-t002:** The image processing pipeline.

Phase	Description	Amount of HS Images	
Radiance	The raw data are processed to radiance (*R*) and its white reference (*W*).	12	Six hyperspectral images, corresponding to combinations of three different light directions and two light wavelength ranges (visible light (VIS) and very near-infrared (VNIR)), and their white references.
Reflectance	From radiance image and its white reference, the reflectance ($R_{refl}$) is calculated as $R_{refl}=\frac{R}{W}.$	6	Six hyperspectral images, corresponding to combinations of three different light directions and two light wavelength ranges (VIS and VNIR).
Combine	The VIS and VNIR images are combined.	3	Three hyperspectral images corresponding to three different light directions.
Albedo and normal	The albedo (*a*) and normal *N* are calculated by Equations (Equation 3)–(Equation 6).	2	One hyperspectral image of the albedo and normal map for the same area.
Depth and albedo	Depth is calculated from *N* by Equations (Equation 7)–(Equation 11).	2	One hyperspectral image of the albedo and the depth map for the same area.
Smoothening	Bregman total variation denoising [29] is applied to the albedo data in the spectral direction. The algorithm is used as implemented in scikit-image version 0.17.2 [30].	2	One hyperspectral image of the albedo and the depth map for the same area.

**Table 3 sensors-22-03420-t003:** Classification report of the pixel classification, slice half model.

	Precision	Recall/Sensitivity	F1-Score	Support
**Healthy skin**	0.63	0.81	0.71	1776
**Benign intradermal nevi**	0.87	0.81	0.84	1697
**Basal cell carsinomas**	0.89	0.76	0.82	2477
**Macro avg.**	0.80	0.79	0.79	5950
**Weighted avg.**	0.81	0.79	0.79	5950

**Table 4 sensors-22-03420-t004:** Leave-one-out (L-O-O) and slice half pixel classification accuracy’s.

Special Feature	Lesion ID	L-O-O	Slice Half	Unique Features
**Typical**	1	0.79	0.94	
**Typical**	9	0.68	0.96	
**Typical**	8	0.62	0.84	
**Typical**	7	0.83	0.90	
**Typical**	10	0.77	0.85	
**Typical**	4	0.70	0.81	
**Typical**	15	0.69	0.79	
**Typical**	14	0.64	0.85	
**Average (typical)**		0.72	0.89	
**Unique**	5	0.23	0.92	Broken skin, dark skin tone, convex surface
**Unique**	3	0.56	0.76	Broken skin, dark skin tone
**Unique**	2	0.25	0.68	Wound or scab on top of lesion
**Unique**	17	0.26	0.94	The lesion and the healthy skin is covered by hair
**Unique**	12	0.003	0.85	Dark skin tone and red-brown lesion tone
**Unique**	11	0.49	0.71	Clear pigment, small lesion
**Unique**	13	0.19	0.81	Two types of lesions: naevus pigmentosus junctionalis
				on right side of the ID
**Unique & difficult**	16	0.16	0.47	Overall fair skin tone, small lesion (2 mm), fair toned and
				uneven delineation
**Unique & difficult**	6	0.38	0.45	Lesion is pigmented, uneven skin tone around lesion
**Average (unique & difficult)**		0.28	0.73	
**Average (all samples)**		0.47	0.81	

## Data Availability

The data presented in this study are not publicly available or available upon request due to ethical and privacy reasons.

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
