# Peer review of "FPI Based Hyperspectral Imager for the Complex Surfaces—Calibration, Illumination and Applications"

_sensors, 2022, doi:10.3390/s22093420_

Round 1

Reviewer 1 Report

The article is complex and overall interesting, for potential clinical significance and completeness of approach: optimization of instrument in response to clinical needs, data acquisition, analysis and classification. 

However, besides some weaknesses, correctly evidenced in discussion chapter and some minor technical points listed below, I find that the main questionable point of the paper is the choice to use CNN to classify a limited number of images without showing and discussing alternative methods. It is common procedure indeed to try several classification approaches and evaluate the relative performances to adapt the right approach to each experimental case and availability/quality of data.

Other minor points:

  • The experiments are described as “clinical pre-test” or “clinical pilot study”: is better to present a single definition to simplify reading
  • I thing that MSI (multispectral imaging), rather than HSI is more correct in this case
  • Paragraph 2.1 line 3: figure 2 instead of figure 1
  • Spectral resolution of the system is stated, but not spatial one (lateral and axial)
  • Also a statement about intensity temporal stability from LED could be useful
  • White and dark acquisition procedure should be briefly indicated
  • Paragraph 2.5.3 not really clear what peak wavelengths 1,2,3 are (combination of FPI and RGB channels?)
  • Graphs and images sometimes lack unit of measure and or dimensions (example fig 11, 13…)
  • No examples of spectra is given, that could be useful to evaluate variability among points and differences among classes
  • The different penetration characteristics of different wavelengths (NIR wavelengths have significant penetration in skin) are not considered in the reconstruction of surface and this could limit the understanding of collected images.

Once addressed these points (and increased literature references to other HSI application to skin diagnosis) I think the article is valuable for publication.

For the next set of experiments, I suggest the author to evaluate the possibility to include in the optical setup elements to limit geometric and chromatic aberrations.

Author Response

Dear reviewer,

Thank you for your valuable comments. They truly helped in making the manuscript better. We provide a point by point response to the comments, and have highlighted the changes we made in the manuscript.

  1. However, besides some weaknesses, correctly evidenced in discussion chapter and some minor technical points listed below, I find that the main questionable point of the paper is the choice to use CNN to classify a limited number of images without showing and discussing alternative methods. It is common procedure indeed to try several classification approaches and evaluate the relative performances to adapt the right approach to each experimental case and availability/quality of data.

The main point of the study is to present the imaging system. To fully compare several classifiers would be enough to justify a full paper, and would also require more data than available for this study. Also, the CNN is a natural choice for the task, as we want to utilize spatial, spectral and 3D characterestics. This is difficult with a versatile tool like CNN, and would require significant amount of preprosessing in random forest or some other traditional machine learning method. In previous research, the CNN was deemed appropriate for task such as this [1,2,3]. We discuss this in the chapter 3.2.3 where the used convolutional neural network is described.

Other minor points:

  1. The experiments are described as “clinical pre-test” or “clinical pilot study”: is better to present a single definition to simplify reading

We opted to use the term pre-test everywhere.

  1. I thing that MSI (multispectral imaging), rather than HSI is more correct in this case

The SICSURFIS HSI is a hyperspectral imager. It can capture from one to thousands of spectral channels depending on its calibration. The number of the spectral channels is indeed relatively low in this study 33 bands), but the device is capable of providing HSI amount of channels with a relatively narrow FWHM ( ~ 10 nm). We included this information to chapter 2.1..  

The high number of dimensions in spectral data can lead to the Hughes phenomenon and redundancy among the samples. These challenges can be avoided computationally for instance by common feature extraction methods. Our solution was to customise the hyperspectral imager to capture only the necessary wavelengths. By selecting the wavelength channels and the corresponding light-emitting diodes (LEDs) to represent the spectral absorption peaks of tissue chromophores from visible light (VIS) to NIR light and using common reflectance calculations, the HS image contains the main diffuse tissue reflectance spectrums, providing a multidimensional view of a lesion with depth information. In this way, we have a computationally effective solution, since the amount of captured data and pre-processing can be limited.

  1. Paragraph 2.1 line 3: figure 2 instead of figure 1

Fixed

  1. Spectral resolution of the system is stated, but not spatial one (lateral and axial)

The imager has a pixel resolution of 1 px ≈ 24 µm x 24 µm. We added this information to chapter 2.1.

As an answer to you, the lateral and axial resolution, the optics of the SICSURFIS HSI is designed with commercial S-mount and C-mount lenses, which provide collimated light beam through the Piezo-actuated FPI tunable filter. The present optical design based on commercial parts provides only a moderate depth of the focus.  We have also studied custom optical systems that would provide a larger depth of the focus and better performance for the generation of the skin surface models (3D).  So the lateral and axial resolution will be taken more carefully attention in the possible next generation devices.

  1. Also a statement about intensity temporal stability from LED could be useful

We added following statement to chapter 2.3.; The temporal stability of LED light source is taken into account by keeping the system on for several minutes before the recording of spectra.  The other way to control the intensity stability is to record white reference images frequently.

  1. White and dark acquisition procedure should be briefly indicated

The dark acquisition was performed by manually placing the imager to a light-blocking holder, seen in Figure 11. The imager was set to capture 40 frames, and the mean is used as a dark reference. The white reference procedure was similar - the imager was placed on a holder against white Teflon. The imager was set to capture with matching LED and wavelength settings, as it captures the HS images. We added this information to chapter 3.1..

  1. Paragraph 2.5.3 not really clear what peak wavelengths 1,2,3 are (combination of FPI and RGB channels?)

We included following explanation to paragraph 2.5.3.;  The peak wavelengths 1, 2 and 3 are determined by the spectral transmission spectrum of the FPI (lower left part of Figure 8). When the FPI transmission curve and R-pixel quantum efficiency are multiplied, we get the Red curve in the center right part of Figure 8. Similarly, we get the Green curve for G-pixels and the Blue curve for the B-pixels in the center right part of Figure 8.

  1. Graphs and images sometimes lack unit of measure Graphs and images sometimes lack unit of measure and or dimensions (example fig 11, 13…)

In figure 11 (now figure 12 due to addition of figure 7) we added the figure size to the caption. In figure 13 (now 14), the absolute values of the surface model are not reliable, as the surface model is at the end of the processing pipeline, and the figure is cleaner without the values visible. We added explinations to figure 17 – 19 captions.

  1. No examples of spectra is given, that could be useful to evaluate variability among points and differences among classes

We added the example spectra to figure 15.

  1. The different penetration characteristics of different wavelengths (NIR wavelengths have significant penetration in skin) are not considered in the reconstruction of surface and this could limit the understanding of collected images.

This has been noted in our conversations withing the team. This is the reason we have used wavelength clearly in the visible range in the analysis. The range of the penetration of the selected wavelengths (0-6 mm) can be seen on chapter 2.1. We added discussion of the matter in the discussion chapter.  

  1. Once addressed these points (and increased literature references to other HSI application to skin diagnosis) I think the article is valuable for publication.

We increased the literature references to contain more HIS skin diagnosis applications and other medical HSI applications

  1. For the next set of experiments, I suggest the author to evaluate the possibility to include in the optical setup elements to limit geometric and chromatic aberrations.

This is an important note. The prototype imager contains commercial lenses. As pointed out in the conclusions, the next-generation device will be designed with a dedicated custom-designed optical system, which can consider these geometric and chromatic aberrations.

References:

  1. Räsänen, J.; Salmivuori, M.; Pölönen, I.; Grönroos, M.; Neittaanmäki, N. Hyperspectral Imaging RevealsSpectral Differences and Can Distinguish Malignant Melanoma from Pigmented Basal Cell Carcinomas: APilot Study.Acta dermato-venereologica2021,101, adv00405.
  2. Ahmad, M.; Shabbir, S.; Roy, S.K.; Hong, D.; Wu, X.; Yao, J.; Khan, A.M.; Mazzara, M.; Distefano, S.;Chanussot, J. Hyperspectral Image Classification-Traditional to Deep Models: A Survey for FutureProspects.IEEE Journal of Selected Topics in Applied Earth Observations and Remote Sensing2021.
  3. Pölönen, I.; Rahkonen, S.; Annala, L.; Neittaanmäki, N. Convolutional neural networks in skin cancerdetection using spatial and spectral domain.Photonics in Dermatology and Plastic Surgery 2019.International Society for Optics and Photonics, 2019, Vol. 10851, p. 108510B.

Reviewer 2 Report

The article present the development of usefull system for obtaining hyperspectral characteritics of human skin/ In general the paper in interest.

Neversless it is desirable to add the comments:

1/  on the temporary characteristics of tunable Fabry-Pero interferometer

2/ the choice of wavelengths  of LED

3/ reduce citation of own works

Author Response

Dear reviewer,

Thank you for your valuable comments. They truly helped in making the manuscript better. We provide a point by point response to the comments, and have highlighted the changes we made in the manuscript.

  1. On the temporary characteristics of tunable Fabry-Pero interferometer

We included this information to paragraph 2.1.; Imager consists of a Piezo-actuated metallic mirror Fabry-Pérot Interferometer (FPI), an RGB sensor and an LED light source. The FPI controls the light’s transmission to the RGB sensor, and the role of the separate long and short pass filters, shown in Figure 2 is to cut the unwanted transmission at not selected orders of the FPI.

The sensor’s basic principle is to provide different spectral layers by changing the FPI air gap [1]. Typically the FPI air gap can be changed to a new value in less than 15 ms i.e. the settling time of the air gap 15 ms.

  1. The choice of wavelengths  of LED

The LEDs were selected to cover the wavelength range of the used fabry-perot interferometer. See figure 7 in the revised manuscript. 

  1. reduce citation of own works

This research is a contribution of three affiliations, each with a long history of skin cancer HSI research. On the sensor point-of-view - the sensor type is unique, and its manufacturer holds the patents of the spectral separator. The operating principles and the detailed mathematical formulas for calculating the spectra can be seen in the previous study.

Since the sensor type is unique, it is valid to use similar methods and compare the results to comparable style data and on the other hand, the previous examples of lesion delineation or lesion type classification are rare.

After re-evaluating the citations, it is noted that some authors might have participated in the previous research, but not all. Since the members of this research consortium have studied this topic with similar style devices and similar patients (fair Nordic skin) over ten years, this research could be seen as a continuation. We balanced the literature references to include more other skin diagnosis HIS and medical applications.

Round 2

Reviewer 1 Report

I appreciate the effort to improve the text of the paper. I still have some doubts about the choices made in the article (starting from CNN to the choice to refer primarly to authors papers in bibliography), but I think that these are part of authors responsibility, so after minor changes (physical dimension bar or indications are still missing in some figures (12, 13)) I suggest the pubblication. 

Author Response

Dear reviewer,

we thank you for your valuable comments and inform you that we included the dimension bars in the final manuscript.

Best regards,

Anna-Maria Raita-Hakola